# Addressing Food Insecurity: A Qualitative Study of Undergraduate Students’ Perceptions of Food Access Resources

**DOI:** 10.3390/nu14173517

**Published:** 2022-08-26

**Authors:** Amanda G. Conrad, Terezie Tolar-Peterson, Antonio J. Gardner, Tianlan Wei, Marion W. Evans

**Affiliations:** 1Department of Food Science, Nutrition and Health Promotion, College of Agriculture and Life Sciences, Mississippi State University, Starkville, MS 39762, USA; 2Department of Counseling, Educational Psychology, and Foundations, College of Education, Mississippi State University, Starkville, MS 39762, USA; 3College of Nursing and Health Professions, University of Southern Mississippi, Hattiesburg, MS 39406, USA

**Keywords:** food insecurity, college students, university resources, resource utilization

## Abstract

Food insecurity has emerged as a leading health care problem in the United States, impacting college students’ health, well-being, and academic performance. The aims of this study were: (1) to assess the prevalence of food insecurity, (2) to identify college students’ perceptions about food access resources, and (3) to explore students’ expressed needs from the university in improving food security status. A mixed-methods approach was used to assess the quantitative and qualitative aspects of the study aims. An online survey to gather demographic information and assess food security status using the 6-item version of the US Household Food Security Scale Module (HFSSM) was administered. Next, qualitative focus groups with subsets of participants was conducted to gain further insight into the perceptions, coping mechanisms, and resource utilization issues related to food insecurity. This study found 34.1% of undergraduate college students to be food insecure and demonstrates that students with a meal plan are less likely to be food insecure (*p* = 0.012; OR = 0.68; 95% CI = 0.489, 0.918). Qualitative data identified key influencers of food insecurity: (1) personal beliefs, (2) life skills, and (3) the university. The results of this study contribute to the literature focused on food insecurity prevalence in college students and presents insight from the college student perspective. Findings may support the development of relevant interventions that are congruent with students’ needs, enhancing resource utilization to increase food security status among college students.

## 1. Introduction

Food insecurity has emerged as a leading health care issue in the United States [1] and impacts college students at a higher rate than found in the general public [2,3]. While there are no national reports on the prevalence of food insecurity among college students, recent literature indicates that about one-third of college students are classified as food insecure [4]. Traditional college-age individuals, those 18–24, have experienced some of the highest rates of unemployment as a result of the COVID-19 pandemic [5] and associations found in the literature between unemployment and food insecurity [6,7] create increased concern about the trajectory of food insecurity among college students given that food insecure students enduring financial hardship are more likely to suspend their collegiate studies than food secure students [8].

Food insecurity is more than feeling hungry. Food insecurity is a broad and multifaceted issue determined by varying levels of influence [9], holding the capacity to create shame and frustration in the students experiencing the circumstance [10,11]. In addition to having higher rates of fair or poor physical health when compared to food secure students [12,13,14], those experiencing food insecurity are more likely to report negative mental health symptoms, such as depression and stress [4,7], which have been identified by students as factors negatively impacting academic potential [15].

To be food secure, students need access to food and food access resources that are believed to be socially acceptable [6,7]. However, even students who have access and are aware of resources, do not always utilize them [16]. The view of normalcy that surrounds being hungry in college can foster reluctance to seek out and utilize resources [10,15].

The goal of the current research is to explore students’ perceptions about food access resources and factors impacting resource utilization, in hopes that the insight gained can lead to the development of effective outreach strategies that link students to solutions that alleviate the problem. By identifying students’ knowledge of interventions, their perceived barriers about utilizing food assistance resources, as well as students’ ideas about potential facilitators to resource utilization, tailored interventions can be developed to match the intended program participant.

## 2. Methods

### 2.1. Study Design

Quantitative and qualitative data collection was conducted to address the study aims. The qualitative method of this study utilized data collected from a subset of 1159 undergraduate college students who were 18 years of age or older, taking classes on the university’s main campus, and who had completed an online food insecurity questionnaire sent to all undergraduate students in the fall of 2019 and spring of 2020. The survey was designed to assess food insecurity prevalence and gather additional descriptive data about the students. Students were classified as food secure or food insecure based on results of the online survey. Students’ university identification codes were used to identify which survey respondents were food secure and which were food insecure. Students were then assigned to focus groups which organized students with similar food security or food insecurity status to minimize stigma and facilitate open discussion [17]. The qualitative design plan of this study had to be altered due to the impact of the COVID-19 pandemic on focus group facilitation. Rather than using face-to-face focus groups as the only method of qualitative data collection, qualitative data was gathered through face-to-face focus groups, virtual focus groups, and online questionnaires. Two face-to-face focus groups were conducted during Fall Semester 2019. Five focus groups were scheduled during Spring Semester 2020, prior to the sudden transition of university classes and operations to distance methods for an undetermined amount of time. The scheduled in-person focus groups transitioned to a virtual focus group format. The transition of the scheduled focus groups to virtual focus groups did not provide equal participation among food secure and food insecure students. Food insecure students had low participation in the virtual focus groups, which led to an underrepresentation of this group. Qualitative data collection was adapted to an online questionnaire with both closed- and open-ended questions based on the focus group prompts. The adaptation of data collection due to a pandemic allowed for increased participation during a time of stay-at-home orders from the university. Components of the online questionnaire had built-in probes so that further information could be obtained based on participants’ responses when indicated. Research on the effectiveness of open-ended questions in web-administered surveys has found potential in the derivation of high-quality responses through “thick, rich, descriptive information from respondents [18]”. Once qualitative data saturation had been reached, recruitment efforts for additional qualitative data participation concluded.

### 2.2. Participants and Recruitment

The research study was approved by the Institutional Review Board (IRB) at Mississippi State University and provided an Exemption Determination (IRB-19-357). All undergraduate students enrolled in traditional classes on campus had been provided the opportunity to participate in a food insecurity survey. A total of 1159 students over the age of 18 completed the food insecurity survey and then the pool was used to recruit participants for the current qualitative study.

Students were recruited to participate in focus groups in the fall semester of 2019 and spring semester of 2020. A recruitment goal for the qualitative aim of this study was determined to be at a point of data saturation, or the point in which potential qualitative themes have been exhausted and no new information is being provided [19,20]. Two focus groups took place in the fall semester, and five focus groups were scheduled for the spring semester. Unfortunately, the COVID-19 pandemic necessitated the cancellation of all in-person focus groups scheduled for the spring. Six virtual focus groups were scheduled and recruited for, but the groups were condensed to three virtual focus groups due to the low number of participant responses. The qualitative data collection method was further adapted to a questionnaire with open-ended questions and sent to all previous survey participants who had not participated in a focus group. A total of two face-to-face focus groups were conducted in the fall 2019 semester, with three participants in each group. A total of three virtual focus groups were conducted in the spring 2020 semester, with two, three, and four participants in each group, respectively. A total of 43 open-ended qualitative questionnaires were completed at the end of the spring 2020 semester. A collective total of 58 students participated in the qualitative arm of this study. Response repetitiveness and data redundancy within the surveys and combined with the focus group responses lead to the establishment of data saturation. Once data saturation was determined, recruitment efforts for additional qualitative data participation was concluded.

### 2.3. Procedures

Quantitative data was obtained to determine prevalence of food insecurity and general student information. Undergraduates were recruited to complete an online survey through a mass email sent from the university with approval by the Vice President of Student Affairs. The recruitment email delivered to all undergraduate students explained the purpose of the research, the type of information included in the survey, the approximate length of time needed to complete the survey, and the voluntary nature of the survey. The email contained statements informing students how to consent to participation in the research by proceeding with the survey link, given their understanding of the research and details of the survey participation.

A pilot focus group was conducted to gain insight on the flow of the focus group questions and to help identify needed revisions to the structure or questions, and for the facilitators to get feedback on their effectiveness in leading the group [17]. A focus group guide was used to conduct the focus groups to ensure consistency, and focus groups were audio recorded for analysis. A co-moderator was present for the first two focus groups and tasked solely with taking notes should the audio equipment fail. The co-moderator was also present in the virtual focus groups. Five open-ended questions, along with prompts and probes for each question were used by the moderator [21]. The questions were prioritized and arranged to allow flow in the focus group discussions and were specifically developed to align with the research objectives. The questions can be found in Table 1. At the end of each focus group, a general summary of the main points that arose were verbally provided by the co-moderator based on the notes taken and participants were allowed to agree or provide insight on the general summary before concluding the focus group [17]. The qualitative data collection method was further adapted to a questionnaire with open-ended questions based on the focus group prompts and probes. The questionnaire was sent to all previous survey participants who had not participated in a focus group. Students completed an informed consent detailing their rights prior to the start of the focus groups or online questionnaire.

### 2.4. Data Analysis

Quantitative analysis was performed by using IBM SPSS Statistics for Windows, Version 27.0. Armonk, NY: IBM Corp. predictive analytics software. Descriptive statistics were determined for demographic data and logistic regression was performed to assess the impact of a set of predictors of food insecurity, should this information be relevant to further discussion identified in the qualitative data. Food security status was calculated based on the method established by the USDA’s scoring methods for the US Household Food Security Survey Module: Six-Item Short Form [22].

NVivo 12 (NVivo version 12.0, QSR International Pty Ltd., 2018) qualitative data analysis software was used. Transcripts of the focus group discussions, as well as participant answers to the open-ended survey questions were read and reread to identify themes via systematic coding [23]. Face-to-face focus groups were transcribed verbatim from audio recordings. Virtual focus groups were transcribed via captioning automated during recording. All transcripts were reviewed and checked for errors. Qualitative data derived from open-ended questions of secondary survey were downloaded into a Microsoft Excel spreadsheet for review and analysis. Each dataset was read and reread to actively seek meaning and patterns in the data. Notes were taken to capture ideas and create initial codes and potential coding themes to use in subsequent phases [24]. Guided by a socio-ecological perspective, thematic analysis was used to provide a systematic framework for coding the qualitative data so that patterns were identified as they relate to the research objectives and an ecological approach to levels of influence [6,7,25,26].

To demonstrate that a rigorous thematic approach was taken to produce insightful analysis of data in relation to the research aims, decisions made about the thematic analysis were explicitly considered [24]. A realist thematic position was used to reflect the reality of food insecurity experienced by college students [24]. This position allows for motivations and experiences to be considered in a straightforward context. To investigate food insecurity as the specific area of interest for this study, a semantic theoretical thematic analysis was utilized, allowing for analysis and coding to be driven by analytic interest in food insecurity in a college student population [24]. Response repetitiveness and data redundancy within the surveys and combined with the focus group responses led to the establishment of data saturation.

Transcript analysis involved systematic coding, informed by the research aims, focus group guides, open-ended questions, as well as the levels of influence which comprised the socio-ecological perspective. To expand the interpretive power of this study, thematic analysis was utilized with a grounded theory approach [27]. Grounded theory coding helps to generate the structure of the analysis by linking the collected data to the emerging patterns being identified [27]. This requires breaking the data down into the basic constituents, defining them as codes, and reviewing the codes to find meaning [27]. Studying the data as it is analyzed can present new ideas to be explored; therefore, multiple rounds of coding can allow for initial ideas to be explored, sorted, and applied back to the data to form interpretations of what is happening in the data [27].

Three rounds of coding were conducted. First, open coding was used to identify initial codes to represent ideas expressed by students. A total of 112 initial codes were named and sorted into potential themes. Relevant codes were collated to produce overarching themes and sub-themes representing various influencers of food security status. The second round of coding used line-by-line coding of overarching themes applied to all transcripts. After the second round of coding, a thematic map was produced to provide a visual representation of themes, as well as structure in identifying the relationship between codes, themes, and level of influence. The levels of influence from an ecological perspective were recognized in the thematic map. A third round of coding was conducted to sort data into themes representing influencers on food security status, with levels of influence at the intrapersonal, interpersonal, and institutional level being considered for their roles as anchors in the upcoming analytic discussion. After the third round of coding, themes were refined to produce the finalized themes.

## 3. Results

Initial codes captured coping mechanisms expressed by both food secure and food insecure students, as well as their perceptions of food access resources. Within the microsystem of each individual student, additional intrapersonal codes were identified which captured the impact of food insecurity on students’ health and everyday activities [28]. Individual beliefs, knowledge, and reactions to food security status were identified. Initial codes also captured the mesosystematic influences of friends, family, living arrangement, and resources accessibility at the interpersonal level. The broader community level of influence was captured in initial coding via discussions involving students expressed needs from the university, as well as students’ ideas around potential changes or involvement by the university that could enhance food access to students.

Several themes representing influencers of food security status became apparent through the quantitative data analysis process. The identified themes were: (1) personal beliefs, (2) life skills, and (3) university. Sub-themes were identified for each of the established themes. For the theme of personal beliefs, identified sub-themes were: (1) resources, (2) what it means to need, and (3) time. For the theme of life skills, the identified sub-themes were: (1) attainment prior to college, (2) implementation, and (3) resourcefulness. For the theme university, the identified sub-themes were: (1) student outreach, (2) interpersonal interactions, and (3) meal plans. The visual relationship of themes and sub-themes for the influences of food security status are presented in the Figure 1 Illustrative quotes representing each theme and sub-theme are in Table 2.

To test the predictive power of independent variables on the dependent variable of food security status, logistic regression was utilized. The model contained nine independent variables based on data obtained in the initial quantitative survey sent to all undergraduate students. The nine independent variables in the model were student classification, ethnicity, gender identity, work status, financial independence, meal plan participation, financial aid recipient, past participation in free- or reduced-school lunch program, and utilization of food assistance resources in the past year. The full model containing all predictors was statistically significant, χ^2^ (9, N = 1091) = 163.07, *p* < 0.001, indicating that the model was able to distinguish between respondents that were food secure from those that were found to be food insecure. The model correctly classified 71.0% of all cases. The independent variable of student meal plans made a statistically significant contribution to the model and is relevant to the qualitative findings. An odds ratio of 0.68 for meal plan was less than 1, indicating that students who have a meal plan were less likely to be food insecure, controlling for other factors in the model.

## 4. Discussion

The literature available on the topic of food insecurity among college students consistently reveals that college students are at risk of food insecurity, and the increased prevalence rates for this group compared to others in their geographic area highlight the need for relevant resources and interventions targeted to college students [2,13,29,30,31,32]. The key findings contribute to the recognition of an increased prevalence of food insecurity among college students and indicate that there are multiple influencers that impact students’ utilization of available resources [7,33].

The findings of this study reveal primary factors expressed by college students that influence food security status. Key influencers are (1) personal beliefs, (2) life skills, and (3) the university. Each influencing entity impacts college students’ ability to cope with food insecurity and reveals more about students’ thoughts regarding food access resources. Some of the themes are similar to associations identified in previous quantitative studies, such as hunger normalcy in college [10], but insight provided through the qualitative aspect of this study enhances what is known about how students feel about resources available to them, and what type of resources will be most relevant and helpful.

Because food insecure students who experience financial hardships are more likely to suspend their studies, university student success and retention programs can benefit from the inclusion of initiatives that normalize the use of food access resources and promote initiatives that lessen the financial burdens associated with meal plan costs [8]. Previous research indicates that “higher education professionals, including student affairs practitioners and others, could be important partners in addressing food insecurity needs” [5]. Findings from this research support a number of multidisciplinary options to enhance college students’ access and utilization of resources which can alleviate food insecurity.

This study supports a previously expressed need “to identify approaches that are both viable and effective in lessening the burden of food insecurity on students” [5]. Viable and effective approaches should be designed for the students who are more likely to be classified as food insecure since these students will be the target audience for the approaches. Previous research has proposed a one-unit undergraduate life skills course as a resource to help alleviate the impact of food insecurity while in college [15]. The influence of life skills on food security status was identified in our study and study participants voiced many ideas that support the benefit of this type of intervention strategy.

This study took a distinct approach to focus on student perceptions of food access resources. Student perceptions and beliefs influence how resources are used to cope with a lack of food. Therefore, an understanding of how students feel about a resource is necessary and can be used by those developing strategies to alleviate college student food insecurity. By identifying the need for interventions which weave strategies representing interpersonal, intrapersonal, and institutional influencers together, the present study offers strategic considerations when planning effective programs relevant to student needs.

### Limitations

A potential limitation of this study is the risk of biasing focus groups to obtain responses that are not necessarily reflective of the participants’ own ideas [34]. Participant selection processes described in the methodology section and the use of a facilitator guide was utilized to minimize bias. Another potential limitation of this study is related to the necessity of focus groups transitioning to a virtual format due to the shut-down of on-campus learning associated with the coronavirus pandemic. Past concerns of virtual focus groups involve the potential elimination of non-computer users [34]. However, due to the support, training, and assistance provided by the university to ensure students had access to virtual learning methods, this limitation was minimized.

## 5. Implications for Research and Practice

The relevance of this study to what we know about the impact of food insecurity on student health and college success is in the application of knowledge gained. Resources to address food insecurity in the college student population must meet the needs of the students and incorporate strategies that will allow for the overcoming of both perceived and actual barriers to resource utilization. Undergraduate students were found to cope with food insecurity through resourceful actions, such as seeking out free food and using resources, as well as by practicing basic life skills, such as budgeting, economically purchasing food, and time management. Many students enter college lacking these basic life skills or do not implement them into their college experience. This influences food security status yet are areas that can be addressed.

### Recommendations

The prevalence of food insecurity in college students is higher than the national average. It is crucial for future research to continue to explore viable intervention strategies that universities can implement which will produce acceptable resources that students will use.

The utilization of meal plans at the university reduces the chance that a student will be food insecure. Meal plans access is a vital option to ensure students have enough food. Making meal plans more affordable and less confusing so that an increased number of students have access to them is one way the university can impact the food security status among college students. Meal plans were mentioned frequently in the qualitative aspect of the study, and students who have a meal plan were found to be less likely to be food insecure. These findings highlight the importance of making meal plans accessible to as many students as possible, as well as a need to reduce barriers in having a meal plan. According to the literature, students who have unlimited access to a meal plan are more likely to be food secure [35]. Students feel that the university can enhance food security on campus by addressing meal plan cost so that food is more affordable for students.

Future research into policies related to meal plan cost and availability will be beneficial in creating policy changes and meal plan programs that enhance food access. The data demonstrates that students with a meal plan are less likely to be food insecure. This finding is supported by statements made by students in focus groups, with one example summarizing the finding in a very succinct manner: “It is definitely easier to get food having a meal plan”. These finding highlight the importance of making meal plans accessible to as many students as possible, as well as a need to reduce barriers in having a meal plan. Data from the qualitative aspect of this study complement the quantitative findings and provide insight from students that a barrier to having a meal plan is the cost. One student indicated that while meals plans are helpful in having enough food, they are too expensive if they are not required. Including meal plan expenses within the cost of tuition is one intervention to consider in attempting to overcoming the financial burden of the increased expense of meal plans. This institutional level intervention would expand the number of students who have access to meal plans, whether the student lives on campus or off campus.

This study also found that students’ resourcefulness and attainment of life skills associated with time and money management influence students’ ability to have enough food. Additionally, the university influences food security status in the way it reaches and interacts with students. Students need to be made aware of resources; therefore, relevant methods used to let students know about resources and the opportunities for students to engage must be present. Students value interpersonal interactions and are more likely to consider using a resource if they learn about it from a person or group that they have a connection with or that they trust. Future research that investigates how students who utilize food access resources heard about the resource can contribute to effective outreach strategies that connect food insecure students to available resources.

Additionally, an intervention strategy that students incorporate into their course schedule, such as a life skills course, will alleviate the expressed issue of not having time to attend programs designed to enhance food access, directly or indirectly. The course setting will allow for discussions surrounding time management, budgeting, efficient utilization of campus meal plans, and economical food purchasing and preparation. Students will be part of a class, where no one is singled out. Resources, and their utilization, can be discussed in a way that normalizes need, reducing stigma. Real conversations can also highlight the prevalence of food insecurity on college campuses to expand perspective and minimize joking about the issue. The need for food access resources and what it means to need those resources are further confused when the level of need is unequal among peers. Peers’ perception of need can fuel feelings of isolation in the struggle with food insecurity, reinforcing the stigma of utilizing available assistance. Jokes about being a “broke college student” lead to uncertainty in students who hear their fellow classmates joke about being poor, but then appear to have money to spend. Encouraging this type of class for students beyond their freshman year will enhance the relevance of the course as students will be less likely to have a meal plan, or an unlimited meal plan, and students will be more likely to be navigating efficient use of more limited meal plans and other food sources. Students discussed the lack of worry they faced when they had a meal plan, but the learning curve needed to learn how to watch their budget when they did not have a meal plan and were balancing grocery shopping with eating out.

## 6. Conclusions

This research provides a voice to food insecurity. It is important for university administrators to know the prevalence of food insecurity on their campus. Knowledge about the local problem will be useful in guiding goals and plans for detection and the connection of students to needed resources that will increase the likelihood of their success, educationally and beyond. While students are often viewed as a general demographic, it is important for individual campuses to understand the challenges faced by their students, and to hear their students’ perspectives of the situation since they are the ones who live the experience. When students provide a voice to food insecurity, universities are provided with the opportunity to listen and continue the conversation with them. This study provides foundational insight into student experiences that impact food security status and can be used to inform future research exploring viable intervention strategies that universities can implement to produce acceptable resources that students will utilize.

## Figures and Tables

**Figure 1 nutrients-14-03517-f001:**
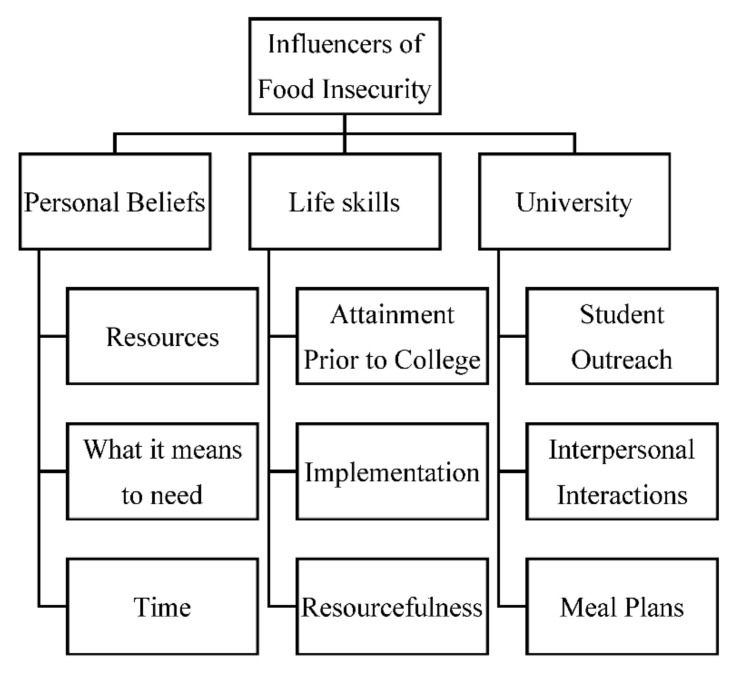
Qualitative results identified influencers of food insecurity status in undergraduate college students.

**Table 1 nutrients-14-03517-t001:** Focus Group Guide: Initial Statements/Question and Subsequent Prompt and Probe.

Initial Statement/Question	Prompt	Probe
Tell me about your food situation in college and if you have enough food or struggle to have enough food.	How do you feel about your food situations?	Do you think your experience is much different or similar to other students’ experiences with food security?
Why do you think having enough food can be a struggle for you or other college students at times?	What are some factors that impact students’ ability to have enough food?	Do you think having enough food is different for college students compared to when someone is not currently attending college?
Tell me how you go about having food to eat as a college student.	What are some things you do to try and have enough food?	Do you seek ways to gain money for food? Do you seek any support from friends or family for food? Are there any activities on campus or in the community you seek out to help you have more food to eat? Do you have ways that you stretch your food dollars?
Tell me about resources that are available to students to help them have enough food to eat.	What do you know about available food pantries in the area? What do you know about the Block-by-Block Meal Program at Mississippi State University? Are you aware of any government food programs that are available to eligible students?	Have you ever used any of these resources? What was your experience like? How did you feel about using the resources? What are some reasons students might not use these resources?
What could be done to better help students have consistent access to healthy food?	Do you feel there is a need for more resources or assistance for students to help them have enough food?	Whose responsibility is this? How would this impact students? What barriers might there be to helping students have adequate food available? Would students be receptive to resources and participate in available programs? Why or why not?

**Table 2 nutrients-14-03517-t002:** Themes, Sub-themes, and Illustrative Quotes for Influencers of Food Insecurity Status in College Students.

Theme	Sub-Theme	Illustrative Quote
Personal Beliefs	Resources	“I think if people just knew about [resources], and if the university talked about food resources as much as they talk about, like, the writing center, yeah. I mean, then if a student is struggling, they remember that information. If the resources are just something that are normal on a college campus and you use it like you use the library, I don’t know, it’s would just be easier to talk about and do”
		“Getting it presented to me by someone with authority is nice, but like getting a personal testimony from students who are like using [a resource] or have done it and they like certify that it is legit and that it helps … that would be a homerun for me”.
Personal Beliefs	What it means to “need”	“There are people who obviously need this more than I do, why should I go get the help?”
		“There’s probably not many external barriers to getting food, but there are definitely still internal barriers, meaning you have to get past any embarrassment and shame and stigma you may feel when going to the pantry to get food”.
		“There is a lot of stigma attached to that sort of thing. I know that when I was a kid I was much more well off and my dad pretty much drilled it into my head that this is not a thing that we need, that we do. We help other people, but we do not require help. It is really difficult to try to get past that mind set, even fifteen years later”.
		“It is hard for students to express it to their friends. It is kind of joked about I guess. Like, ‘Oh, college kids, we are just living off of Ramen’. … And it is discouraging to even mention to your other friends, when one minute everyone is joking about how broke they are, and saying ‘I’m on the Ramen diet’, or just making the broke college kid jokes, when the next they are doing things that are like, things they would clearly not do if they were broke”.
		“So, it’s just kind of becomes one of those things people joke about and don’t really think about those who are truly affected, like, when the words literally mean it … right?”
Personal Beliefs	Time	“I mean, your classes end at five and you have class from ten to five, five days a week. But then you have two hundred pages of reading for each class each week, and then you have like two ten-page essays due at the end of the semester and then three other essays due. So, then you don’t have free time. You read on your phone the book you are supposed to be reading, and you are sort of cooking whatever dinner you are supposed to have. But you don’t have free time to prep anything, or work more, or even use some of the stuff out there that is supposed to be helping you have food”.
		“And sometimes [resource organizers] are not considerate of other people’s time, but they are insisting that this is a great thing that you guys need to do”.
		“So, in college, I’m definitely like, time is money”.
Life Skills	Attainment prior to college	“So, like I would go grocery shopping with [my mom]. I would see how she would like, buy things, how she would, like use, like, rice, five different ways. You know, so definitely. I learned that from my mom growing up”.
		“But my friends will buy a frozen pizza and that is pretty much is. But I know that you can buy, like, frozen chicken breast, and a five-pound bag of potatoes and tons of canned vegetables. You just have to know how to shop on a budget. It is a mindset. But it does take knowing how to cook … and stretch it … just stuff like that. The frozen pizza is one meal, and it is gone”.
		“Access to resources and how to cook or grocery shop or purchase food on a budget is needed—students don’t know how to do that stuff. It is a shift and they’ve never dealt with it before. But now, they have to”.
Life Skills	Implementation	“I keep track of how many block meals I have and how many weeks I have left and then divide. So, I divide and am like, okay, I can use this many block meals per week. So, like every week I make sure I know. So, I count beforehand and plan”.
		“I have friends who say they are struggling with food, but I know they blew their meal plan at the beginning just buying junk food …”
		“We would find ways to maximize the meal plan, but that would use up a lot of time figuring it all out”.
		“It is very hard to get used to a meal plan. I remember my freshman year, I was like, so these places you can go after nine [o’clock], these places you can’t, you can do the things where you can go to these buffet places and use your meal plan again at Chick-fil-a, and then use two block meals, It was just…hard. You have to figure it out to make it work”.
		“It takes time to get food and prepare the food we want”.
Life Skills	Resourcefulness	“Well, for me, I have time to cook for myself, I just don’t have money. It is a lot cheaper to stretch a can of beef stew and a cup of rice out for two days than it is to have three square meals a day”.
		“Rice. It is a God send. It stretches so well, and it lasts forever so it won’t go bad. “The first freshman week to campus is great! There is food everywhere!”
		“Sometimes they will have these events and they will just say, like, ‘free food’ at the bottom. So, sometimes I will go.”
		“I know a lot of churches have free food. Really, any of the churches in [town] Starkville. Just check and you can usually get free food from there”.
		“The first freshman week on campus is great. There is food everywhere!”
		“I sneak food from work. Crackers and stale pieces of bread that would have been thrown away. I’m not proud of this”.
University	Student Outreach	“I think the university can ask more about students needing food. Send an email and ask and just give them a number to call for those who need it. Make it easy. And nobody has to know. Everyone got the email. Nobody knows if you call”.
		“I think that many students do not know about how to ask for help when they don’t have enough food. Therefore, I think the university should have flyers or more ways for students to ask for help in a more private manner so that they don’t feel embarrassed”
		“I think emails from the university are helpful. But not just from them. The organizations I am involved in, they send out follow-up emails or either a group message notification saying, ‘Hey, this is what they just sent out. Make sure you read it, share it with others”.
University	Personal Interactions	“Me, I just have learned about what is offered on campus because we just talk about it in the different clubs I am in”.
		“It helps when the programs can be talked about in classes. It helps to try to bring the real world in with, like, what we are learning. So, like, talking about these programs, and it may be because of the classes I am taking, but classes are a good place because students are already involved in class and have to hear about it. So, students who are involved learn about things, but hit the things that students are involved in, like, for sure”.
		“This is about our time and what we are paying for. I mean, make [telling about a resource] a part of the discussion where we can use it”.
		“For me, a personal connection makes [a resource] mean more”.
		“I know if I had my band director tell me something, I’m like, ‘Oh, I should probably do this’. But if my meat science teacher is telling me, I’m probably not going to listen”.
		“Sometimes the university does programs to make themselves look good. This is just my hot take on it. But I think, sometimes if they aren’t being considerate of other people’s time, it seems fake … I think maybe it is your RA’s [residence hall assistant] responsibility or those who are leaders in your social circles, or people who are supposed to be there for you”.
University	Meal Plans	“It is definitely easier to get food having a meal plan. I was more likely to go eat because it had already been paid for and it was one less thing to think about”.
		“Meal plans do help you have access to food. But they are too expensive when you are not required to have it”.
		“The meal plans are too expensive. I am paying my own way through college and deciding if I wanted to pay that much for a meal plan was hard to choose”.
		“So, like, the Student Association, they do this Block by Block meal thing. So, you can donate a meal plan which I think is so interesting. Because we say all the time that we have an overabundance with the unlimited meal plan, yet they do this thing for people who are, like, food insecure. And I am like, this is really cool, but I am also like, it means there is something inefficient with our meal plans if they are always asking us to donate”.
		“Students should be able to donate more of their unused meals since they already paid for it”.
		“They don’t let you swipe a meal for a friend and share, but they tell you to donate, but you can only donate a certain amount”.
		“Instead of making Block by Block optional, the university should automatically take the leftover blocks and donate them to food insecure students the next semester”.
		“Being limited to only two block meals that can be donated makes people feel as though they aren’t making a difference and, some of my friends have been turned off from donating”.

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
