# Peer review of "Addressing Food Insecurity: A Qualitative Study of Undergraduate Students’ Perceptions of Food Access Resources"

_nutrients, 2022, doi:10.3390/nu14173517_

Round 1
Reviewer 1 Report
Reviewer comments
The Abstract is too long. The authors should consider reducing it to 250 words.
Line 28-29: What were the p values and 95% CIs?
The authors should consider introducing a “Conclusion” in the Abstract
Keywords: The authors should consider adding more keywords
Methods
The authors should state the number of subset of University student
Line 73: How were the students grouped as those with “food security” and those “food insecurity?” The authors should state how that was done
Line 75: How was the alteration done?
Line 90: Delete “from”
I have a challenge with the sample size that was used. How did the authors settle on 58? That is inadequate.
Discussion
The discussion of this work is highly inadequate. The authors used just 2 references. Several studies have been conducted on food security which the authors have compared their findings with.
The authors should create a section title for “Conclusion” and “Recommendation”
General comments:
In the Abstract, the authors indicated some of the outcomes of interest that were strong predictors of food insecurity. However, reading through the manuscript, the authors showed no statistical analysis including Regression analysis on any of those outcomes. This is a weakness of this work.
I also have a challenge with the use of a sample size of 58 for this work which the authors indicated in the Method. There is no scientific justification to that.

Reviewer 2 Report
In the manuscript „Addressing Food Insecurity: A Qualitative Study of Undergraduate Students’ Perceptions of Food Access Resources” the Authors tried to examine students’ perceptions about food access resources and factors impacting resource utilization. A mixed-methods approach was used to assess the quantitative and qualitative aspects of the study aims. The Authors conducted the online survey and the qualitative focus groups with subsets of participants. The quantitative part of the study included 1159 students. Moreover, total of 58 students participated in the qualitative part of this study. The study is an important contribution to science due to its subject including food insecurity.
Generally, the manuscript provides valuable information. The Authors very clearly described methods. Moreover, the Authors have got approval from the Institutional Review Board at Mississippi State University. The Authors used proper procedure in data analysis. However, I have some questions and remark.
Line 32-35;
The Authors wrote that the results contribute to the literature focused on food insecurity and help to fill in gaps in understanding food insecurity as well as allow developing relevant interventions. The Authors should be more precise and detailed, referring directly to their study.
Discussion;
In addition, the discussion is not conducted in a comprehensive manner. There is a lack of reference to many of the results. The discussion should be a scientific dialogue. It should be primarily an exchange of thoughts. The Authors should better refine the discussion.
Round 2
Reviewer 1 Report
The authors have thoroughly revised the manuscript and I can see that the quality of the content has improved significantly. The depth in statistical analysis provided at 224-237 is very essential. Thank you for undertaking this revision.